# Research on Factors Affecting SMEs' Credit Risk Based on Blockchain-Driven Supply Chain Finance

**Ping Xiao** [1,2,*], **Mad Ithnin bin Salleh** [1] and **Jieling Cheng** [2]

1. Faculty of Management and Economics, Universiti Pendidikan Sultan Idris, Tanjong Malim 35900, Malaysia
2. College of Mathematics and Finance, Hunan University of Humanities, Science and Technology, Loudi 417000, China
* Correspondence: pingxiao_88@163.com

**Abstract:** The development of blockchain-driven supply chain finance aimed to solve the financing problems of SMEs. However, credit risk is expanded, and even transmitted to the whole supply chain, due to their connection, so that it becomes more difficult to effectively identify the credit risk of SMEs. The purpose of this paper was to examine the factors affecting SMEs' credit risk in the mode of block-chain-driven supply chain finance. This research proposed an entropy weight method to construct independent variables and used logistic regression to examine whether the financing enterprises, core enterprises, assets position under financing, blockchain platform, and supply chain operation have significant impacts on credit risk. The panel data, originating from CSMAR on fifty-six quoted SMEs, included eight core enterprises and twenty-six blockchain enterprises, between 2016 and 2020. The results showed that the financing enterprises, core enterprises, asset position under fi-nance, blockchain platform, and supply chain operation have significant impacts on SMEs' credit risk when the confidence level is 90%. The financial status of financing enterprises can reflect the credit status of SMEs. Core enterprises give credit guarantees to SMEs, and the business transactions between SMEs and core enterprises affect the credit risk through the asset position under financing. Meanwhile, blockchain platforms can solve the problem of the information asymmetry of the participating enterprises in supply chain operations. At the same time, the supply chain operation is also an important factor affecting the credit risk. This conclusion provides a reference for the ap-plication of blockchains in supply chains, to reduce the credit risk. At the same time, the selected indicators were more comprehensive, which provided a strong basis for the subsequent construc-tion of a credit risk assessment model using key factors.

**Keywords:** credit risk; blockchain-driven supply chain finance; SMEs; entropy weight method; logistic regression



## 1. Introduction

The financing difficulties of small and medium-sized enterprises (SMEs) have always been a concern in China, due to weak financial strength, high operational risk, and low financial transparency [1]. SMEs often struggle to obtain financial support through equity financing, due to an information system dominated by state-owned capital [2] and a high credit risk hidden in the further development of accounts receivable financing [3]. At the same time, information asymmetry leads to repeated pledges, empty pledges, and other risk events, which seriously hinder the mutual trust of financing parties and bring difficulties to financing [4,5].

The the supply chain finance (SCF) model aims to solve the financing problem of SMEs [6]. SCF was originally defined as a short-term financing solution [7–9]. However, the credit status of core enterprises and SMEs is enlarged due to the connection of the supply chain, and the credit risk is also expanded, and even transmitted to the whole supply chain; in particular, it becomes more difficult to effectively identify the credit

risk of financing enterprises due to there being more participants in blockchain-driven supply chain finance. How to combine the risk sources, strengthen the corresponding risk management, and effectively control the risk is the key to the success of supply chain finance business [10]. Therefore, a flexible supply chain system with a high level of information integration from production, warehousing, logistics, distribution, to retail is gradually established, which cultivates fertile "soil" for the application of blockchain technology in this scenario [4]. Blockchain is an emerging technology that can solve the problems of asymmetric information and low financing efficiency in supply chain finance [11]. The traditional financial mode can no longer adapt to the rapid development of society, due to the lack of trust among related parties. To solve this problem, blockchain is one of the effective solutions that could establish a fully decentralized, reliable, and sustainable financial platform, without influences from any institutions or individuals [12].

Previous studies on SCF mainly focused on the credit risk assessment system, but failed to determine the key factors affecting the credit risk of SCF [1,13–17]. Studying the key factors that affect the credit risk of SMEs can enable capital providers to effectively understand SMEs and provide targeted financial services [18,19]. Research has shown that credit risk, as a special risk, spreads in SCF networks and may affect other participants [20–22]. Therefore, it becomes more difficult to effectively identify the credit risk of enterprises, because credit risk is a systemic risk in complex financing [23]. Therefore, there is a need to study this association among the factors of credit risk in a blockchain-driven supply chain finance mode. The main objective of this research was to find influencing factors on SME credit risk, using a logistic regression method to examine whether financing enterprises, core enterprises, assets position under financing, the blockchain platform, and supply chain operation have effects on credit risk, which is the basis for credit risk assessment. This study constructed a credit risk assessment index, then used logistic regression to test the factors affecting SMEs' credit risk.

Compared to previous studies [15,24–27], a wider range of indicators were selected in our study. Not only quantitative indicators, but also qualitative indicators, were selected. This study pays attention not only to the participating enterprises in the blockchain-driven supply chain, but also to the macroeconomic environment and national policies. The established index system is more extensive, which lays a foundation for building a credit risk assessment model.

The remainder of this study is structured as follows: Section 2 is a literature review, with an extensive explanation of the credit risk, financing enterprises, core enterprises, assets position under finance, blockchain platform, supply chain operation, and hypotheses development. Section 3 is based on the proposed methodology. The results are comprehensively discussed in Section 4, and the conclusions, containing the theoretical and practical implications, limitations, recommendations, and future work, are summarized in Section 5.

## 2. Literature Review

SMEs upstream and downstream of a supply chain are on the demand side of creditor's rights financing, which are called financing enterprises, and the core enterprises are creditor's right enterprises and anti-guarantee creditor's right enterprises [28]. The role of financial characteristics in the credit risk assessment of financing enterprises is considered, such as the operating profit margin, current ratio, and total asset turnover, which are important predictors of the default of SMEs [24]. Altman and Sabato [29] proved that cash, total assets, earnings before tax, interest payments, retained earnings, short-term liabilities, and equity are the main influencing factors of financing enterprise credit risk. Zhang, H. et al [17] selected financial indicators that can reflect the profitability, operation ability, solvency, and development ability of financing enterprises, to reflect the financing ability of SMEs.

The core enterprises are the guarantors of SMEs, and the credit level of these counterparties has become an important concern in the risk management of financial institutions [13]. At the same time, as the core of the supply chain, the operation of core enterprises

can also affect the profit margin of SMEs in supply chains [30]. Zhu, Y. et al. [14] selected some financial indexes to represent core enterprises, such as the credit rating, quick ratio, turnover of total capital, and profit margin on the sales. In addition, enterprise scale, asset-liability ratio, credit-rating, and profitability were selected to represent the risk related to the core enterprise [16].

The asset position under financing can be reflected in the accounts receivable, inventories, and prepayments of SMEs. When SMEs default, banks can sell off their assets under the financing, to make up for the losses caused by the credit risk of the SMEs [1]. The receivables turnover ratio and inventory turnover ratio were selected to represent the assets position under financing, which are related to financing enterprise [31], because the accounts receivable are generated between the financing enterprise and the core enterprise [16].

A blockchain platform is an organic combination of distributed ledger technology, point-to-point technology, asymmetric encryption technology, intelligent contract technology, consensus mechanism, and a series of existing mature technologies [32]. In order to protect the privacy of sensitive data in the supply chain finance scenario, a new method using homomorphic encryption in blockchain was proposed [4]. Chen et al. [12] implemented a blockchain-driven supply chain finance platform, which aimed to ensure trust between shareholders and reduce the financing costs of the automobile retail industry. By recording the data flow between different types of institutions in the financing process, a blockchain can effectively improve the transparency of financial information and the traceability of data [33].

The diffusion of credit risk in a supply chain financial network will have serious consequences. As one of the important risks of supply chain finance, credit risk is infectious, and leads to the spread of credit risk in the supply chain economy [34]. For example, banks, core enterprises, SMEs, and the risk intermediary subsystem interact to form the SCF credit risk system in a supply chain mode [35]. Zhao [34] examined the impact of key factors, such as general financing proportion, recovery time, network structure, and network scale, on the diffusion process of credit risk, and focused on the diffusion and stability of credit risk in a supply chain financial network. It is reasonable to give priority to macroeconomic factors, such as GDP growth rate, M2 growth rate, and interest rate [13].

Some scholars [15,16,26,33,36,37] have studied the factors that affect the credit risk of SMEs, based on the supply chain finance model below:

### 2.1. Financing Enterprises and Credit Risk

The financing enterprises and the problems of the supply chain itself may result in SMEs' inability to fulfil their legal obligations and FIs facing SME credit risk in SCF [1]. Once the financing enterprises in the supply chain are faced with insufficient market demand, the decline of sales performance will increase their credit risk, and they may default on loans to commercial banks and the core enterprises in the supply chain. The contagion effect of SME credit risk on the core enterprises will have a significant negative impact on their operation and development, and then affect the stable operation of the whole supply chain [38]. It is considered that there are four main factors influencing the credit risk of SMEs: profitability, structure, liquidity, and operating conditions and characteristics [25]. The profitability, operational ability, and solvency of financing enterprises may lead to credit risk [26]. From the above research, it can be seen that financing enterprises are the factors that affect SMEs' credit risk.

### 2.2. Core Enterprises and Credit Risk

SMEs are either upstream or downstream, which depends on the development of core enterprises. Therefore, the good credit status of core enterprises is an important guarantee for the survival and development of SMEs in the supply chain [39]. Financing enterprises deal with core enterprises and bear some problematic credit risks through the cooperation of core enterprises [17]. Financial institutions do not directly participate in the

operation of supply chain finance and mainly rely on the credit level and operational status of core enterprises to carry out credit rating. Therefore, the credit level of core enterprises determines the credit risk of financing enterprises [37]. Wang Zongrun et al. [40] found that under the supply chain finance model, SMEs can improve their credit status with the help of core enterprise credit. When the credit capacity of core enterprises is insufficient, they may not be able to rely on their own credit line to provide guarantees for financing enterprises applying for loans [16].

The credit of the core enterprise is given a great weight in the credit evaluation of the whole supply chain [15]. The credit risk of the core enterprise is the main source of the credit risk of the whole supply chain, and an increase in the number of core enterprises will improve the credit risk immunity of the whole system [41]. Tian Kun, Zhuang Xintian, and Zhao Wanting [27] established a credit risk assessment model for SMEs using the solvency, profitability, and related factors of core enterprises through factor analysis and logistic regression; the result showed that the current ratio, asset liability ratio, return on net assets, and operating net interest rate of core enterprises have a significant impact on the credit level of SMEs. From the above research, it can be seen that core enterprises play an important role in credit risk.

### 2.3. Assets Position under Financing and Credit Risk

Supply chain finance leads to the mutual infection and superposition of credit risk between upstream and downstream enterprises [42]. The relationship between SMEs upstream and downstream can be reflected in the asset position under financing, such as accounts receivable financing and inventory financing [15]. Accounts receivable financing allows upstream enterprises to obtain short-term credit loans from banks. This not only helps solve the short-term capital needs of financing enterprises and promotes the healthy and stable development of SMEs, but also helps with the continuous and efficient operation of the whole supply chain and reduces credit risk [43]. From the above research, it can be seen that the asset position under financing has a significant impact on the SMEs' credit risk.

### 2.4. Blockchain Platform and Credit Risk

Credit risk is non-normally distributed and complex because of the characteristics of information asymmetry, dissemination, and the difficultly of monitoring [36,43]. Some scholars believe that financial information is the main factor affecting the credit risk of SMEs [14]. In the blockchain platform, the flow of information is transparent and accessible between the principal and agent, which could reduce credit risk [44], because the nodes in the blockchain are verified using digital signature technology, and a cross enterprise trust mechanism is built through the alliance chain [4]. As decentralized technology, blockchain technology can improve the speed of business processes without face-to-face trust, and reduce the financing costs and risks more effectively [45]. Blockchain technology can also detect and warn of the credit risk caused by enterprise fraud and tampering with data [33]. With blockchain-driven supply chain finance (BCT-SCF), a win–win situation among supply chain participant is possible, which is helpful for the supply chain in choosing a more efficient financing channel [46]. From the above research, it can be seen that blockchain platforms have an effect on reducing SMEs' credit risk.

### 2.5. Supply Chain Operation and Credit Risk

Supply chain finance also has environmental risk, financing risk, information technology risk, human resource risk, and basic structure risk, which can damage the survival and solvency of financing enterprises [47], and finally, this turns into the credit risk of the enterprise. As a bridge to the supply chain, supply chain financing integrates the financing process of all members in the supply chain and improves the value of the supply chain, which is different from the traditional manner of financing. There is a credit risk propagation among the interconnected enterprises [34]. Therefore, in supply chain financing, the credit risk of SMEs is not only affected by SMEs themselves, but also affected by supply

chain financing, such as the financial and non-financial status of enterprises, supply chain operation status, and the characteristics of pledge objects [14]. Any risk factor affecting a single enterprise in the region throughout the supply chain will affect other participants. Therefore, the supply chain itself has great uncertainty [37].

More and more researchers have incorporated non-financial information into their research on factors affecting credit risk, such as the macroeconomic environment, core enterprise credit, supply chain level, market trends, company credit record, and management team quality [48]. Lu et al [28] showed that the supply chain can help service providers reduce financial risk, while supply chain operation is also one of the influencing factors on credit risk. It was concluded that the operating profit margin, total asset turnover, and current ratio of enterprises have a great impact on the industry situation of enterprises, and the strength of supply chain cooperation has an impact on the credit risk of supply chain finance.

From the previous research and Table 1, it can be seen that all participants in the blockchain-driven supply chain finance mode have an impact on the credit risk of financing enterprises [49], such as core enterprises [38], blockchain platform [50], supply chain operation [14,51], and asset positions under financing [15]. How to effectively evaluate credit risk is the primary task to reduce credit risk. In particular, credit risk factors are analyzed to obtain a risk assessment model of blockchain-driven supply chain finance. This study empirically investigated whether the financing enterprises, core enterprises, asset position under financing, blockchain platform, and supply chain operations have effects on credit risk in this mode.

**Table 1.** Comparison between previous research and this research.

| Authors | Previous Research | This Research |
|---------|-------------------|---------------|
| Zhu et al., 2016 [49]; Yang, Y., et al., 2021 [26]; Wen, F, Bi, et al., 2021 [16] | Financial data were comprehensively selected, but they were all quantitative data, with a lack of qualitative data. | In addition to quantitative indicators, there are qualitative indicators, such as credit status and the relationship strength of supply chains. |
| Zhu, Y., et al., 2019 [14]; Aslam et al., 2021 [44]; Du et al., 2020 [4]; Choi T.M., et al., 2019 [45]; Yan Zhengya, 2021 [33] | They clarified the impact of information asymmetry on credit risk and proposed the impact of blockchain on credit risk, but did not quantitatively analyze the impact of the blockchain platform on the credit risk of SMEs. | Selected the blockchain platform enterprises and relevant data to quantitatively study whether the blockchain platform had an impact on the credit risk of SMEs. |
| Zhao, Z. B., et al., 2018 [34]; Zhu, Y., et al., 2019 [14]; Su and Lu, 2015 [48] | They paid attention to the macro-economic environment, but did not pay attention to national policies. | This study paid attention, not only to the participating enterprises in the blockchain-driven supply chain, but also to the macroeconomic environment and national policies, such as sustainability. |

## 3. Research Methodology

In order to avoid the gap in financial status between different industries, this study focused on SMEs in the same industry. The postal savings bank of China found that the manufacturing industry is in the primary position among primary industries. Therefore, this study located its sample in the manufacturing industry, which allowed avoiding the errors resulting from the differences between different industries. We collected data from the China Economic and financial research database (CSMAR) and the financial statements of listed companies, recommending a minimum sample size of 90 enterprises (56 SMEs, 8 core enterprises, 26 blockchain platforms) to test a logistic regression model in the period 2016–2020; we excluded the following:

Enterprises that do not participate in the supply chain model. As the SMEs studied in this study are based on a blockchain-driven supply chain finance mode, the collected

enterprises had to be in the supply chain mode to meet the research objectives, so the enterprises without a supply chain relationship were deleted

Firms have missing variables in the data. The empirical results cannot truly reflect the relationship between variables when there is a missing value, so the missing values were deleted [52].

In the process of data analysis, duplicates were dropped. In the process of data collection, duplicate data is introduced when importing with software. Duplicate data have no significance for an empirical analysis [52], so duplicate data must be deleted.

### 3.1. Procedure of Sampling

This study integrated the participating enterprises under the blockchain-driven supply chain finance mode, then established an index system according to the basic structural requirements of the above credit risk assessment index system. As most indicators come directly from China's CSMAR database [17], the financial data in the sample were mainly from the CSMAR database, and the non-financial data were obtained through the company's annual report and related news reports.

The blockchain platform provides an opportunity to connect various participants in the entire supply chain network [12]. Therefore, this study started with the blockchain platform to collect samples. Although the financial indicators cannot fully reflect the situation of SMEs, the financial situation is indeed the main indirect indicator for measuring the ability of enterprises from all aspects. Only a good financial situation can support the loan repayment behavior of SMEs [17]. SMEs deal with core enterprises and bear some credit risks through the cooperation of core enterprises. Therefore, the relevant information of core enterprises should also be considered in the evaluation index [17]. Sampling procedures were consulted from Zhu You [1], as follows:

(1) Selecting samples from the blockchain platform.

We found 26 blockchain platform enterprises from a white paper on the development of blockchain financial applications [53], which had frequent business with manufacturing industry related SMEs in the supply chain finance mode. By browsing the official websites of the 26 blockchain platform enterprises and collecting cooperation information with SMEs, 56 SMEs were found to have business with the 26 blockchain platform enterprises.

(2) Screening of SMEs.

As for how to divide risky SMEs and non-risky SMEs, the standard adopted in this paper was special treatment enterprises (ST). If a small and medium-sized enterprise is a ST listed company, this means that it is a small- or medium-sized enterprise with an abnormal financial situation and losses in successive accounting years, which is facing the risk of delisting. Therefore, these kinds of small- and medium-sized enterprises are classified as risky SMEs. On the contrary, non-ST Listed SMEs are defined as non-risky SMEs [1]. From the screening, the 56 small and medium-sized enterprises included 11 ST companies and 45 non-ST companies, so a set of risky enterprises, including 11 groups of samples, and a set of non-risky enterprises, including 45 groups of samples, were constructed.

(3) Screening of core enterprises.

In order to better reflect the characteristics of the supply chain, this study located the core enterprises cooperating with SMEs and the supply chain, where each small and medium-sized enterprise was located by searching the enterprise's official website, the Oriental Wealth, the company's annual report, and other relevant information; and a total of eight main board enterprises were selected. Among the 56 SMEs, each enterprise had actual business dealings with a selected large enterprise, and eight large enterprises had the typical characteristics of core enterprises in the supply chain.

### 3.2. Measures of Variables

The dependent variable was credit risk, and the independent variables were financing enterprises, core enterprises, assets position under financing, blockchain platform, and

supply chain operation. The variables were measured using a sub-index derived from a previous literature review [14,15,26,27,48].

### 3.3. Measures of Credit Risk

This study divided the research objects of credit risk ($CR_i$) into two categories: one is risky SMEs, the other is non-risky SMEs. The value of the dependent variable represents whether the signal released by a small- or medium-sized enterprise is a high credit risk or no credit risk. In order to facilitate the construction of the model, the signal released by risky SMEs is defined as negative, and the value is 0. The signal released by non-risky SMEs is defined as positive, and the value is 1. As for how to divide risky SMEs and non-risky SMEs, the standard adopted in this paper was ST [1] (Zhu You, 2016).

$$CR_i = \begin{cases} 1, non - risky \\ 0, risky \end{cases} \tag{1}$$

### 3.4. Measures of Independent Variables

According to previous research, this study used the entropy weight method. The financing enterprise ($FE_{it}$) is measured from profitability ($Profitability_{it}$), growth ability ($Growth_{it}$), operation ability ($Operation_{it}$), and debt paying ability ($Debt_{it}$). The core enterprise ($CE_{it}$) is measured from profitability ($Profitability_{it}$), operation ability ($Operation_{it}$), and debt paying ability ($Debt_{it}$). The assets position under financing ($AF_{it}$) consist of receivable turnover ratio ($Recievable_{it}$) and inventory turnover ratio ($Inventory_{it}$). The blockchain platform ($BC_{it}$) is measured from profitability ($Profitability_{it}$), operational ability ($Operation_{it}$), and debt paying ability ($Debt_{it}$). The supply chain operation ($SC_{it}$) contains the relationship strength of the supply chain ($Strength_{it}$), macro environment ($Macro_{it}$), and sustainability ($Sustainability_{it}$).

$$FE_{it} = \alpha_1 Profitability_{it} + \alpha_2 Growth_{it} + \alpha_3 Operation_{it} + \alpha_4 Debt_{it}$$
$$CE_{it} = \alpha_1 Profitability_{it} + \alpha_2 Operation_{it} + \alpha_3 Debt_{it}$$
$$AF_{it} = \alpha_1 Recievable_{it} + \alpha_2 Inventory_{it}$$
$$BC_{it} = \alpha_1 Profitability_{it} + \alpha_2 Operation_{it} + \alpha_3 Debt_{it}$$
$$SC_{it} = \alpha_1 Strength_{it} + \alpha_2 Macro_{it} + \alpha_3 Sustainability_{it}$$

$\alpha_i$ is the weight calculated using the entropy method, $i$ represents enterprise, and $t$ represents year; thus, $it$ represents the sample of an enterprise in a certain year.

### 3.5. Data Collection

From existing research results, the indicators of the credit risk under the supply chain financial environment can be divided into two aspects: one is the enterprise's own risk factors, which are basically the same as the traditional assessment system. On the other hand, this is a correlation analysis, which mainly includes the status of core enterprises, blockchain platform enterprises, supply chain operation, etc. The data were collected from the CSMAR database and financial statements on the official website of the enterprise.

Based on the credit risk assessment index system constructed by scholars at home and abroad, and the operational process of the supply chain financial business model driven by the blockchain platform, this study selected indicators to evaluate the credit risk of SMEs from the following dimensions: This study confirmed 27 credit risk sub-indicators, which were used as explanatory variables to build the model. The system divided the indicators into five categories.

There are three levels of indicators in total, and the independent variables are in the first-level index. The weighted average method was used to convert the three-level indicators into two-level indicators. On account of the influence of each two-level indicator on the primary indicators being different [54], this study used the entropy method to convert the two-level index into primary indicators. According to the indicators of Table 2,

the weighted average method was used to consolidate the indicators, as shown in the index formula from Table 2.

**Table 2.** Independent variables involved in credit risk assessment.

| Primary Indicators | Two-Level Indicators | Index Formula | Symbol | Three-Level Indicators | Equation |
|---|---|---|---|---|---|
| FE | Profitability | Profitability = $(X_1 + X_2)/2$ | $X_1$ | Net interest rate on total assets | Net profit/total average assets |
| | | | $X_2$ | Operating profit ratio | Operating profit/ operating income |
| | Growth | Growth = $(X_3 + X_4)/2$ | $X_3$ | Growth rate of total assets | The degree of growth of total assets relative to the previous year |
| | | | $X_4$ | Net profit growth rate | The degree of growth of net profit this year compared to the previous year |
| | Operation | Operation = $(X_5 + X_6)/2$ | $X_5$ | Current asset turnover | Main business income/average total current assets |
| | | | $X_6$ | Turnover of assets | Operating income/total assets |
| | Debt | Debt = $(X_7 + X_8 + X_9 + X_{10})/4$ | $X_7$ | Current ratio | Current assets/ current liabilities |
| | | | $X_8$ | Quick ratio | Quick assets/current liabilities |
| | | | $X_9$ | Cash ratio | Cash/current liabilities |
| | | | $X_{10}$ | Asset–liability ratio | Total liabilities/total assets |
| CE | Profitability | Profitability = $(X_{11} + X_{12})/2$ | $X_{11}$ | Net interest rate on total assets | Net profit/total average assets |
| | | | $X_{12}$ | Operating profit ratio | Operating profit/ operating income |
| | Operation | Operation = $(X_{13} + X_{14})/2$ | $X_{13}$ | Turnover of assets | Operating income/total assets |
| | | | $X_{14}$ | Credit status | Defect in internal control |
| | Debt | Debt = $(X_{15} + X_{16})/2$ | $X_{15}$ | Quick ratio | Quick assets/current liabilities |
| | | | $X_{16}$ | Asset–liability ratio | Total liabilities/ total assets $\times$ 100% |
| AF | | Receivable = $X_{17}$ | $X_{17}$ | Receivable turnover ratio | Operating income/accounts receivable closing balance |
| | | Inventory = $X_{18}$ | $X_{18}$ | Inventory turnover ratio | Current cost of sales/average inventory balance |
| BC | Profitability | Profitability = $(X_{19} + X_{20})/2$ | $X_{19}$ | Operating profit ratio | Operating profit/ operating income |
| | | | $X_{20}$ | Main revenue growth rate | The degree of the main growth this year compared to the previous year |
| | Operation | Operation = $(X_{21} + X_{22})/2$ | $X_{21}$ | Turnover of assets | Operating income/ total assets |
| | | | $X_{22}$ | Credit status | Credit rating/guarantee ratio |
| | Debt | Debt = $(X_{23} + X_{24})/2$ | $X_{23}$ | Quick ratio | Quick assets/current liabilities |
| | | | $X_{24}$ | Asset–liability ratio | Total liabilities/ total assets $\times$ 100% |
| SC | Strength | Strength = $X_{25}$ | $X_{25}$ | Relationship strength of supply chain | Whether financing enterprises and core enterprises have long-term supply and marketing contracts |
| | Macro | Macro = $X_{26}$ | $X_{26}$ | Macro environment | Industry total assets net profit margin |
| | Sustainability | Sustainability = $X_{27}$ | $X_{27}$ | Sustainability | Refer to GRI:1; No refer to GRI:0 |

In the analysis of financial statements, the three-level indicator is the subdivision of the two-level indicator. Each subdivision indicator has the same effect on the two-level

indicator, so the weight is the same, and the weighted average method is adopted. However, the two-level indicator is divided according to the financial management ability, which will have different impacts on risks [54], so the entropy weight method was adopted.

From Table 2, profitability indicates the profitability of the enterprises, growth represents the growth ability of the enterprises, operation is the operational ability, and debt designates the debt paying ability of the enterprises.

*3.6. Data Preparation*

3.6.1. Data Matrix

This study made a comprehensive analysis of 27 three-level indicators of 280 evaluated objects, indicating m = 280. Assuming that the evaluated objects are $P = (P_1, P_2, \cdots\cdots, P_M)$, the evaluation indicators are $G = (G_1, G_2, \cdots\cdots, G_n)$, and the evaluation objects $P_i$ record the sample $x_{ij}(i = 1, 2, \cdots, 280; j = 1, 2, \cdots, 27)$ of indicators $G_j$, which $x_{ij}$ represents the relevant $j$ indicator of the $i$ company in a certain year, the original matrix is formed as follows:

$$I = \begin{pmatrix} & G_1 & G_2 & \cdots & G_n \\ P_1 & x_{11} & x_{12} & \cdots & x_{1n} \\ P_2 & x_{21} & x_{22} & \cdots & x_{2n} \\ \vdots & \vdots & \vdots & \vdots & \vdots \\ P_m & x_{m1} & x_{m2} & \cdots & x_{mn} \end{pmatrix} \tag{2}$$

where $P_i$ represents the supply chain participating enterprises from 2016 to 2020 (m = 280), and $G_j$ represents the basic index (n = 27).

3.6.2. Data Preprocessing

In the comprehensive evaluation, there are different types and dimensions among the evaluation indexes. Therefore, in order to eliminate the impact of these differences, it is necessary to deal with these evaluation indexes as dimensionless. The commonly used linear dimensionless methods include the standardization method, extreme value method, linear proportion method, normalization method, vector norm method, and efficacy coefficient method. In this paper, the entropy extreme value method was more applicable for the preprocessing of data [55].

The extreme value processing method is a linear transformation of the original data. Let the sum be the maximum $M_j$ and minimum $m_j$ values of attribute $A$, respectively, and map an original value $x_{ij}$ of $A$ into the value $x_{ij}^*$ in the interval [0,1] using the extreme value processing method.

If the evaluation index $x_{ij}$ is positive

$$x_{ij}^* = \frac{x_{ij} - m_j}{M_j - m_j}(i = 1, 2, \cdots, n; j = 1, 2, \cdots, m) \tag{3}$$

If the evaluation index $x_{ij}$ is inverse

$$x_{ij}^* = \frac{M_j - x_{ij}}{M_j - m_j}(i = 1, 2, \cdots, n; j = 1, 2, \cdots, m) \tag{4}$$

*3.7. Entropy Weight Method*

The entropy method is a mathematical method used to judge the dispersion degree of an index. The greater the degree of dispersion, the greater the impact of the index on the comprehensive evaluation. The greater the dispersion of the value of an index $G_j$ and $X_{ij}$, the higher the degree of disorder of the information, indicating that the indicators can provide more information for the final evaluation goal. According to the value of $p_{ij}$, the

$p_{ij} = \frac{X_{ij}}{\sum_{i=1}^{m} X_{ij}} (0 \leq p_{ij} \leq 1)$ entropy $e_j$ of the $j$ index can be obtained. Where $X_{ij}$ is the $i$ evaluation object under the $j$ indicator.

$$e_j = -\frac{1}{\ln(m)} \sum_{i=1}^{m} p_{ij} \ln(p_{ij}) \tag{5}$$

According to the value of $e_j$, this research calculated the difference coefficient and entropy weight $w_j = \frac{g_j}{n - \sum e_j} (j = 1, 2, \cdots, n)$, where the difference coefficient $g_j$ of the $j$ index is $g_j = 1 - e_j$. Since the denominator is a fixed value, the greater the difference coefficient, the greater the amount of information provided by the indicators, and the greater the entropy weight. According to the entropy weight of each indicator layer divided by the cumulative entropy weight of the indicator of the system layer, the weight $\alpha$ of each index layer is obtained. Finally, the primary indicators (independent variable) represented by the two-level indicators are calculated as follows:

$$FE_{it} = 0.00352 Profitability_{it} + 0.00329 Growth_{it} + 0.7683 Operation_{it} + 0.2249 Debt_{it} \tag{6}$$

$$CE_{it} = 0.07 Profitability_{it} + 0.54 Operation_{it} + 0.39 Debt_{it} \tag{7}$$

$$AF_{it} = 0.2487 Receivable_{it} + 0.7513 Inventory_{it} \tag{8}$$

$$BC_{it} = 0.027 Profitability_{it} + 0.263 Operation_{it} + 0.71 Debt_{it} \tag{9}$$

$$SC_{it} = (Strength_{it} + Macro_{it} + Sustainability_{it})/3 \tag{10}$$

All the data are brought into the above formula to obtain the samples of $FE_{it}$, $CE_{it}$, $AF_{it}$, $BC_{it}$, $SC_{it}$. In order to verify whether the independent variables have a significant impact on $CR_{it}$, this paper used binary logistic regression for the empirical analysis.

Table 3 shows a comparison between the predicted value and the actual data, with 0.5 as the dividing line between credit risk and non-credit risk. Among these, "Observed" represented actual data, and "Predicted" represented predicted values. CR represents a SMEs' credit risk. It can be seen from the table that 12 observation objects with credit risk were correctly predicted, and the accuracy rate was only 21.8%. At the same time, 223 observation objects without credit risk were correctly predicted, and the accuracy rate was 99.1%. The overall correct judgment rate was 83.9%. The overall prediction accuracy was good, but the prediction accuracy for enterprises with credit risk was poor, at only 21.8%. In the following section, we tested whether the independent variables had a significant impact on the dependent variables through logistical regression.

**Table 3.** Classification Table.

| | Observed | | | Predicted | | |
| --- | --- | --- | --- | --- | --- | --- |
| | | | | CR | | Percentage Correct |
| | | | | 0 | 1 | |
| Step 1 | CR | | 0 | 12 | 43 | 21.8 |
| | | | 1 | 2 | 223 | 99.1 |
| | Overall Percentage | | | | | 83.9 |

The cut value is 0.500.

Table 4 shows the relevant statistics of the variables in the logistical regression model. Among them, FE represents financing enterprises, CE represents core enterprises, AF represents assets position under financing, BC represents blockchain platform, and SC represents supply chain operations. Sig. represents the *p*-value result of the significance test, in which the p-value of FE, AF, BC, SC, and constant term test is less than 0.05, and the p-value of CE test is greater than 0.05. The results showed that the financing enterprises, assets position under finance, blockchain platform, and supply chain operation have a

significant impact on credit risk (Sig. < 0.05), but the core enterprise has no significant statistical significance, indicating that the core enterprise has no direct impact on credit risk. However, when the confidence level is 90% (Sig. < 0.1), all the independent variables have a significant impact on credit risk. From the above results, it can be seen that financing enterprises, core enterprises, asset position under financing, blockchain platform, and supply chain operation are indeed factors affecting credit risk.

**Table 4.** Logistic Regression.

|  |  | B | S.E. | Wald | df | Sig. | Exp(B) |
|---|---|---|---|---|---|---|---|
| | FE | −6.381 | 2.750 | 5.384 | 1 | 0.020 | 0.002 |
| | CE | −1.473 | 0.880 | 2.804 | 1 | 0.094 | 0.229 |
| Step 1 [a] | AF | −7.672 | 3.198 | 5.754 | 1 | 0.016 | 0.000 |
| | BC | 7.751 | 2.141 | 13.101 | 1 | 0.000 | 2322.910 |
| | SC | 2.450 | 1.178 | 4.323 | 1 | 0.038 | 11.585 |
| | Constant | 0.371 | 0.717 | 0.268 | 1 | 0.605 | 1.449 |

[a] Variable(s) entered on step 1: FE, CE, AF, BC, SC.

## 4. Discussion

According to the results of the logistic regression in Table 4, the financing enterprises have a significant effect on the SMEs' credit risk. Although banks want to provide credit to these enterprises, they often refuse loans because SMEs have a small scale, insufficient collateral, difficult production and operation, and a poor ability to resist economic fluctuations, of which the financial indicators can best reflect the above situation [56]. The financial situation of financing enterprises is the direct reason for whether SMEs have credit risk. Although the financial indicators cannot fully reflect the situation of SMEs, the financial situation is indeed the core indirect indicator for measuring the ability of enterprises from all aspects. Only good financial ability can help support the loan repayment behavior of SMEs [17]. Therefore, this study selected the profitability, growth ability, and debt paying and operation ability of SMEs, which can fully reflect the financial situation of the financing enterprises [1,15,26,27].

Differently from the traditional financing mode, supply chain finance uses the reputation, status, and economic strength of core enterprises in the supply chain to provide guarantees for the financing of SMEs, so as to help SMEs in the supply chain obtain loans from financial institutions. Therefore, core enterprises play an important role in supply chain finance. The impact on the supply chain financial credit risk of SMEs cannot be underestimated, which is embodied in the credit status of core enterprises and their own strength; while the upstream and downstream enterprises in the supply chain share the capital risk of the core enterprises, but do not obtain the credit support of the core enterprises [57,58]. Therefore, the logistic regression test results showed that the core enterprises have no significant statistical significance when the confidence level is 95% (Sig. < 0.05), so the status of core enterprises has no direct impact on credit risk. However, when the confidence level is 90% (Sig. < 0.1), core enterprises have a significant impact on credit risk.

Another reason is that although SMEs and core enterprises are upstream and downstream enterprises, the SMEs depend on the development of core enterprises. However, the relationship between them is mainly reflected in the asset positions under finance in the supply chain model [15], such as accounts receivable, accounts payable, and prepayments [16,31]. When SMEs default, the bank can sell the assets under the financing to make up for the losses caused by the default of SMEs [1]. For example, accounts receivable financing refers to that SMEs upstream and downstream of the supply chain, which can borrow from financial institutions with undue accounts receivable, on the premise that the core enterprises of the supply chain promise to pay [28]. Therefore, the asset position under financing has a significant impact on credit risk, according to the logistic regression.

In the traditional supply chain financial model, the problem of information asymmetry in the financial market makes commercial banks unable to fully trust the core enterprises, and they were still required to spend a lot of money and energy on risk control links [36,43]. For example, a blockchain platform can invest human or material resources to check the enterprise credit data or verify the authenticity of transaction data; as the blockchain's technical data cannot be tampered with, it can ensure the authenticity of the information and data obtained by commercial banks, without needing additional costs to carry out end-of-end risk control activities, and this can help financial institutions identify the authenticity of the data. In addition, time delays, human errors, and cost factors can be eliminated, and fraud or product theft can be prevented [59]. Therefore, it can be seen from Table 4 that a blockchain platform has a significant effect on credit risk.

It can be seen from the results that a supply chain operation has a significant impact on credit risk, indicating that long-term supply and marketing contracts between financing enterprises and core enterprises play an important role in credit risk. The stability of supply chain relationships is a long-term and regular supply chain transaction in the supply chain, and the key factors of a stable supply chain transaction are stable enterprise financial status and a good level of industry development; whereby the enterprise factors and the future of industry development can represent the stability of supply chains long-term, and stable trade relations represent good cooperative relations [17], and the industry total asset net profit margin represents the profitability of the whole industry; meanwhile, sustainable development conforms to national advocacy policies.

## 5. Conclusions

From the perspective of blockchain driven supply chain finance, this study estimated whether the financing enterprises, core enterprises, asset position under financing, blockchain platform, and supply chain operation have a significant impact on SMEs' credit risk. First, according to the theoretical research [8,12], data indicators were constructed from the SMEs' subject, SMEs' debt, and the blockchain platform. Second, the weighted average method was used to convert the three-level indicators into two-level indicators, and then the entropy weight method was used to convert the two-level indicators into primary indicators (independent variables). Finally, logistic regression was used to test the relationship between independent variables and dependent variables. The results show that the financing enterprises, core enterprise asset position under finance, blockchain platform, and supply chain operation have significant impacts on credit risk when the confidence level is 90% (Sig. < 0.1). As SMEs are the direct beneficiaries of supply chain finance, their own situation is the basis of credit risk. This study mainly reflected their situation through the financial situation of SMEs. Generally speaking, the possibility of an enterprise repaying a loan depends largely on the financial situation of the enterprise. The better the financial performance, the greater the possibility of repaying the debt on time and the lower the credit risk. Specifically, this is reflected in the enterprise's profitability, debt repayment, operational ability, and growth ability to obtain cash. The financial status of financing enterprises can reflect the credit status of SMEs.

Differently from the traditional financing mode, because of the strong reputation, status, and economic strength of the core enterprises, supply chain finance relies on the core enterprises to provide guarantees for the financing of SMEs, to help SMEs in the supply chain obtain loans from financial institutions. Therefore, when the confidence level is 90% (Sig. < 0.1), core enterprises have a significant impact on credit risk. Accounts receivable financing refers to the SMEs upstream and downstream of the supply chain being able to borrow from financial institutions with undue accounts receivable, on the premise that the core enterprises of the supply chain promise to pay [28]. Therefore, asset position under financing has a significant effect on credit risk. Blockchain can accelerate the dissemination of information in the supply chain and improve the authority of information [60]. Therefore, it can be seen from the logistic regression results that the blockchain platform has a significant effect on credit risk. Meanwhile, supply chain operations have a significant

impact on credit risk, indicating that long-term supply and marketing contracts between financing enterprises and core enterprises play an important role in credit risk.

Our work contributes to research and practice. First, we cited the blockchain platform as an independent variable, to study whether blockchain has an impact on the credit risk of SMEs, which provides a reference for the application of blockchain in supply chains. Second, we investigated the impact of non-financial indicators on the credit risk of SMEs, and compared to previous studies [1,15,16,26,36], the selected indicators were more comprehensive, which provided a strong basis for the subsequent construction of credit risk assessment model through key factors. Third, according to the results, the blockchain platform can solve the problem of the information asymmetry of participating enterprises in supply chain operations, which provides a guide for reducing the credit risk of supply chains.

However, this study also has some limitations, which can be addressed with further research. First, this study only used the data of SMEs in the manufacturing industry to verify the factors affecting credit risk. Furthermore, the sample size was not large enough, on account of only enterprises engaged in manufacturing being selected. Future research could collect data from different industries, to enrich the model and theory proposed, and so that the results would be more general. In the current SCF practice, the Type I error is too high, indicating that credit risky enterprises are wrongly judged as non-risky enterprises; that is, loans are issued to the wrong borrowing enterprises, which is not conducive to the development of supply chain finance.

**Author Contributions:** Formal analysis, Data curation, Writing—original draft, P.X.; Supervision, M.I.b.S.; Funding acquisition, J.C. All authors have read and agreed to the published version of the manuscript.

**Funding:** This research was funded by 2021 innovation training program for college students in Hunan Province: Credit risk identification and model construction of sustainable supply chain finance [XJT (2021) No. 20, 3690] and mathematics research fund project of School of mathematics and Finance (2020SXJJ06).

**Data Availability Statement:** China Stock Market & Accounting Research Database is available at https://www.gtarsc.com/ (accessed on 10 August 2022).

**Acknowledgments:** The authors would like to thank the College of Mathematics and Finance, Hunan University of Humanities, Science and Technology for supporting this work.

**Conflicts of Interest:** The authors declare no conflict of interest.

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
