# Peer review of "Research on Factors Affecting SMEs’ Credit Risk Based on Blockchain-Driven Supply Chain Finance"

_information, doi:10.3390/info13100455_

Round 1

Reviewer 1 Report (New Reviewer)

Based on the blockchain-driven supply chain finance, the factors affecting the credit risk of small and medium-sized enterprises can be analyzed in this work. However, this paper has the following problems:

1.        In this work, financing enterprises, core enterprises, assets position under financing, blockchain platform and supply chain operation are analyzed to understand the impacts on credit risk. These selected objects do not appear to belong to the same type and level, why do you choose the analysis objects like these?

2.        As you say, “From the above research, it can be seen that blockchain platform are the factors leading to SMEs’ credit risk”. The blockchain platform should have the positive impact on the credit of SMEs. Why do you think that the blockchain platform can lead to SMEs’ credit risk?

3.        What’s the classification criteria for SMEs and core enterprises in this work?

4.        Shorter variable names in the formulas are recommended, which can make your paper more academic. Moreover, the equations in Table 2 should be organized better to make the equations more unified and understandable.

5.        Why are the weights of the two-level index determined by the entropy weight method and why are the weights of the three-level index determined by the weighted average method in this work?

6.        Table 3 seems redundant because the weighted average method can be better described by some formulas and explanations combined with Table 2.

7.        In Section 3.6.1, 280 evaluated objects are selected. However, the sample size of 90 enterprises is selected in Section 3. Why the sample quantity does not correspond?

8.        The extreme value processing method and primary indicators in Section 3.6.2 should be written as the formulas and are numbered.

9.        The full word of ST should be given in Section 3.

10.    More analyses are necessary for the results in Table 3 and Table 4. The reason corresponding to the results should be further analyzed. Moreover, the results should be separated into sections.

11.    In Section 4, the discussion should be described combined with the research data of Table 3 and Table 4, which is not another literature review.

12.    In this work, some abbreviations, such as supply chain finance (SCF), are repeatedly defined. Please check it.

13.    There are two Table 3 in this work. Moreover, the Table 3 Classification Table need to be redesigned to make the table clearer.

14.    Some sentences are difficult for readers to understand, such as “Identifying the key factors affecting the credit risk of SMEs could effectively identify SMEs and provide targeted financial services”. Please the full text carefully, and make the sentences easier to understand.

15.    There are some grammar errors in this work, such as the absence of some articles and conjunctions.

Author Response

Thank you for your valuable comments on the article. I have revised the article according to your comments. Please see the attachment. Thank you very much!

Reviewer 2 Report (Previous Reviewer 1)

1.       “the selected indicators are more comprehensive”. “More comprehensive” can be only in comparison with?

2.       The resource “Yan Zhengya, 2021. Research on supply chain financial model innovation from the perspective of blockchain and Internet of things. Xinjiang social science, 2: 47-50.” is not available on the internet for the international community. It cannot be cited.

Author Response

Thank you for your valuable comments on the article. I have revised the article according to your comment. Please see the attachment.

Reviewer 3 Report (Previous Reviewer 2)

All my previous comments are addressed. So, I would propose accepting the paper as it is.

Author Response

Thank you for your suggestions and approval of the article.

Reviewer 4 Report (Previous Reviewer 3)

Agree

Author Response

Thank you for your suggestions and approval of the article.

Round 2

Reviewer 1 Report (New Reviewer)

This manuscript has been revised according to the review form.

This manuscript is a resubmission of an earlier submission. The following is a list of the peer review reports and author responses from that submission.

Round 1

Reviewer 1 Report

1.      No consistency in writing the key term “SMEs”. In the title, it is written like “Smes”.

2.      “the credit risk of financing enterprises due to the complex mode.” It is not clear “the complex mode” of what.

3.      “Here we propose”. “Here” is the noise word.

4.      “The panel data, originating from CSMAR on fifty-six quoted SMEs, eight core enterprises and twenty-six blockchain enterprises between 2016 and 2020. ”. Where is a verb in this statement?

5.      “The results verify that financing enterprises”. Why “verify”? I suppose “conform”.

6.      “the selected indicators are more comprehensive”. “More comprehensive” can be only in comparison with?

7.      “The financing difficulty of small and medium-sized enterprises (SMEs) (Falkner & Hiebl, 2015)”. I do not see meaning in this reference.

8.      “To solve this problem. blockchain” – typo.

9.      “Previous studies on supply chain finance (SCF) mainly focused on the credit risk assessment system, but failed to determine the key factors affecting the credit risk of SCF.”. References are needed.

10.   “Altman and Sabato (2007)”. Wrong year.

11.   “a series of existing mature technologies (Boroujerdi, 2015)”. Such reference does not exist.

12.   References are ordered in no order. It is confusing.

13.   Who is writing “Ieee”?!

14.   “the traceability of data (Yan Zhengya, 2021).”. Wrong year. The manuscript is full of inaccuracies. I’m not going to trace all of them. Such carelessness of authors shows total negligence to the preparation quality of the manuscript. Such a manuscript does not deserve to be considered for publication.

15.   “this paper selected the profitability, growth ability, debt paying and operation ability of SMEs, which can fully reflect the financial situation of financing enterprises.” How this selection can be compared with the variables selected by other researchers?

Reviewer 2 Report

1.  The abstract is very long. Shorten it. The conclusion part in the abstract should be shortened.

2.  Explain the contribution of the paper clearly in the introduction.

3.  Explain blockchain and supply chain clearly in the introduction.

4.  Literature review  is not clear enough. Make a table describing advantages, disadvantages of the existing research and how your work improves it.

5.   Number all equations and explain them clearly. 

6.  Explanation of the proposed solution is dataset specific. First explain your solution which is general and then later it can be made dataset specific.

7. The results  are not presented clearly. 

Reviewer 3 Report

The goal of this paper is to study the factors influencing the credit risk for SME in the case of blockchain-driven supply chain finance. The authors propose a number of 5 factors and resp. a number of 27 third-level factors for assessment. Then the logistic regression method is used to check the influence on the credit risk. An example including sets of block chain platforms, core enterprises and financing SME is demonstrated.

The idea of the paper is interesting. The main contribution of the paper is applied one, namely: identifying the factors about credit risk and studying their influence.

Main remarks about technical quality of the paper.

A number of paragraphs duplicated in meaning in the conclusion (see discussion). See page 14 from the beginning till “Our work contributes …”. Also, at several places in the text there are phrases like “this paper makes, this paper integrates, this study …” etc. It will be ok to change the statements. The references are not in alphabetical order or on first citation.

- Please, use capital letter for “SME” in the title

- G = (G1, G2, …, Gn) not Pn (see page 10)